# Stress Field Approach for Prediction of End Concrete Cover Separation in RC Beams Strengthened with FRP Reinforcement

**DOI:** 10.3390/polym14050988

**Published:** 2022-02-28

**Authors:** Binbin Zhou, Ruo-Yang Wu, Shiping Yin

**Affiliations:** 1Jiangsu Key Laboratory of Environmental Impact and Structural Safety in Engineering, School of Mechanics & Civil Engineering, China University of Mining and Technology, Xuzhou 221116, China; 2State Key Laboratory for Geomechanics & Deep Underground Engineering, China University of Mining and Technology, Xuzhou 221116, China; 3Wilson and Company, South Jordan, UT 84096, USA

**Keywords:** end concrete cover separation, stress field approach, cracked concrete, failure strength, dowel action, concrete splitting

## Abstract

End concrete cover separation is one of the most common failure modes for RC beams strengthened with external FRP reinforcement. The premature failure mode significantly restricts the application of FRP materials and could incur serious safety problems. In this paper, an innovative stress field-based analytical approach is proposed to assess the failure strength of end concrete cover separation and the conventional plane-section analysis is extended to evaluate the corresponding carrying capacity of FRP-strengthened RC beams. First, the dowel action of reinforcement and the induced concrete splitting, reflecting the interaction between concrete, steel and FRP, are considered in establishing the geometrical relationships of stress field for cracked concrete block. Then, the cracking angle and innovative failure criterion, considering the arrangement of steel and FRP reinforcement and cracking status of concrete and its softening effect, are derived to predict the occurrence of concrete cover separation and related mixed modes of debonding failure. Subsequently, an extended sectional analytical approach, in which the components of effective tensile strain of FRP resulted from flexural and shear actions are both considered, is presented to evaluate the carrying capacity of strengthened beams. Finally, the proposed calculational model is effectively validated by experimental results available in the literature.

## 1. Introduction

Due to the pronounced advantages, such as high strength, light weight, electromagnetic transparency, non-corrosive, and nonconductive properties, externally bonded (EB) fiber-reinforced polymer (FRP) and near-surface-mounted (NSM) FRP have become the prevailing techniques over the last three decades for flexural strengthening of existing reinforced concrete (RC) members [1,2,3,4]. Extensive experimental and analytical research has been performed to investigate the structural performance of FRP-strengthened RC members and to assess the retrofitting efficiency. Accordingly, numerous study findings indicated that premature reinforcement debonding failure restricts the sufficient application of FRP materials and furthermore, the brittle failure could incur serious safety problems of RC members or structures [5,6]. According to the failure mechanism, debonding failure can be divided into interfacial debonding (ID) that happens at or near a bi-material interface and concrete cover separation (CCS) that occurs along the level of internal tensile steel reinforcement. Moreover, debonding failure can be also categorized into reinforcement end debonding and intermediate crack-induced debonding in terms of failure location [6,7], which are schematically shown in Figure 1.

Concrete cover separation at the end of FRP reinforcement has been found to be the common failure mode in the retrofitting techniques using EB and NSM FRP methods and has obtained increasing research attention [8,9,10,11,12,13]. Plenty of experimental investigations have been carried out to explore the failure mechanism and influential factors of concrete cover separation, and to assess its failure strength [8,9,10,11,12,13,14,15,16,17,18,19,20,21,22,23,24]. For example, Garden et al. [15] experimentally investigated the influence of FRP plate anchorage length on carrying capacity and failure mode of strengthened RC beams. The strengthened beams were found to fail in concrete cover separation under low shear span–depth ratios. Yao et al. [16] implemented the comprehensive experimental investigations of FRP-plated RC beams containing a variety of geometrical and material parameters. Experimental findings indicated that most of strengthened RC beams failed in concrete cover separation at FRP plate end and that the failure load due to debonding closely correlated with the stiffness of FRP bonded plate and concrete cover. Aprile et al. [17,18] experimentally and analytically investigated the crack spacing, crack pattern, failure strength of EB FRP-strengthened RC beams under uniform load conditions, and found that crack spacing is a key parameter to assess the failure strength of concrete cover separation. Teng et al. [10] and De Lorenzis et al. [4,19] summarized the available debonding failure patterns including concrete cover separation, crack configurations, and the formation locations of failure in experimental campaigns of RC beams strengthened with NSM FRP strips. Barros et al. [12,13] and Bilotta et al. [20] experimentally compared the retrofitting efficiency of RC beams strengthened with NSM FRP strips and EB FRP plates. Findings showed that the RC beams strengthened with NSM FRP strips were more prone to concrete cover separation due to the higher bond efficiency and that the concrete cover separation was often accompanied by the formation of diagonal cracks. Czaderski [21] experimentally investigated the crack configurations of separated concrete cover of EB FRP-strengthened RC beams, and analytically derived the failure strength of concrete cover separation using strut-and-tie model. Sabzi et al. [22,23] explored the influence of arrangement details of tensile steel reinforcement on concrete cover separation of RC beams strengthened by FRP sheets through experimental investigations with the major variables of reinforcement ratio and reinforcement diameter. Experimental results demonstrated that the highly reinforced concrete beams were more vulnerable to concrete cover separation compared to the moderately and lightly reinforced concrete beams and that increasing number of reinforcement and reducing reinforcement diameter would depress the occurrence of concrete cover separation. Sharaky et al. [24] experimentally verified the obvious effect of interaction between NSM FRP and steel reinforcement on the failure modes of strengthened RC beams through adjusting the arrangement of NSM FRP. It was found that concrete cover separation could be delayed or prevented by deepening the location of NSM FRP. Through experimental investigations of RC beams strengthened with FRP sheets, Al-Saawani et al. [25] suggested that concrete cover separation was the major failure mode for the strengthened RC beams with shear span–depth ratio less than 3.0.

Many analytical models and numerical techniques have been also presented to evaluate the failure strength of concrete cover separation [2,3,4,5,6,7,8,9,10,11,12,13,14,15,16,17,18,19,20,21,22,23,24,25,26,27,28,29,30,31,32,33,34,35,36,37,38,39,40,41,42,43,44,45]. The concrete tooth model [29,30] is one of the well-known analytical models, and furthermore, has been extended by numerous researchers [27,31,32,37,38]. For the concrete tooth model, the concrete block between two adjacent inclined cracks at the end of FRP reinforcement was modeled as a cantilever under the action of horizontal shear stress at the tensile side of the strengthened RC beam, as shown in Figure 2. Debonding was considered to initiate as the tensile stress at the roof of the concrete block, resulted from the shear stress applied by FRP reinforcement, reached the tensile strength $f_{ct}$ of concrete. Accordingly, the fracture moment $M_{ct}$ applied to the concrete block is expressed by Equation (1)
(1)$$M_{ct}=T_{f}c_{n}\;$$
where $T_{f}$ is the resultant force of the shear stress applied to the concrete block; and $c_{n}$ is the thickness of concrete cover.

In addition, the fracture moment $M_{ct}$ of the concrete block can be also solved by Equations (2) and (3) according to the assumption of concrete in elastic state:(2)$$M_{ct}=f_{ct}J_{th}\frac{2}{S_{r}}\;$$
(3)$$J_{th}=\frac{b_{w}S_{r}^{3}}{12}\;$$
where $J_{th}$ is the second moment of area of concrete block; $b_{w}$ is the width of beam cross-section; and $S_{r}$ is the crack spacing of strengthened RC beams.

Consequently, the resultant force $T_{f}$ of tensile FRP reinforcement resulting in concrete cover separation is assessed by Equation (4):(4)$$T_{f}=\frac{f_{ct}b_{w}S_{r}^{2}}{6c_{n}}\;$$ The corresponding effective tensile strain of FRP reinforcement is expressed by Equation (5) [26,27,28,29,30,31,32]:(5)$$\varepsilon_{fe}=\frac{f_{ct}b_{w}S_{r}^{2}}{6E_{f}A_{f}c_{n}}\;$$
where $E_{f}$ is the elastic modulus of FRP reinforcement; and $A_{f}$ is the area of FRP reinforcement.

Equation (4) illustrates that the effective tensile strain of FRP reinforcement figured out by concrete tooth model is highly sensitive to crack spacing; therefore, crack spacing is the most critical factor to predict the failure strength of concrete cover separation. Consequently, the modified concrete tooth models by refining the calculation of crack spacing were subsequently presented [8,32]. According to the detailed retrofitting techniques and configurations, the concrete tooth model was further extended for the strengthened RC members with NSM FRP bars or strips [26]. Although the other influential factors such as cracking status of concrete interacted with surrounding steel and FRP reinforcement, the softening effect of compressive concrete, and arrangement details of steel and FRP reinforcement were not incorporated into the mechanical model. Furthermore, the hypothesis of concrete in elastic state generally results in the pronounced discrepancies between predictions and experimental results [33,34]. In addition, the more complex mixed modes of debonding failure seems not to be accurately predicted by the model. The similar defects were existed in some other well-known analytical models presented by Roberts [40], Malek et al. [41], Oehlers [42], Jansze et al. [43], and Ziraba et al. [44] which are based on the mechanical analysis of local point [45].

Concrete tooth model and the other mentioned analytical models have been so widely employed due to its simplicity and convenience in analysis and design. However, the great simplifications render the pronounced drawbacks in accuracy and application scopes. Thus, the more sophisticated numerical techniques have been used to improve the accuracy of prediction of concrete cover separation. Hawileh et al. [35] presented an advanced finite-element (FE) model to assess the global deformation development of RC beams strengthened with NSM FRP rods. Bond behavior of steel and NSM FRP reinforcement with adjacent concrete surface was comprehensively considered in the simulations of concrete cover separation. Based on nonlinear fracture mechanics, Camata et al. [36] numerically predicted the concrete cover separation occurring at the FRP plate end and at the midspan of strengthened RC beams. In simulations, the crack configurations were predefined, and the cracking process was described by an interface crack model. Zhang et al. [37] presented a discrete crack model in the numerical analysis of FRP-plated RC beams that failed in concrete cover separation. The cracking and failure criterion of concrete and the bond law between concrete and FRP were incorporated in the simulations of concrete cover separation. Similarly, to precisely predict the occurrence of concrete cover separation of FRP-plated RC beams, Radfar et al. [38] implemented the energetic criterion into finite-element analysis, in which the crack configurations were predefined. Besides FRP reinforcement and concrete, which are the major focus of the aforementioned studies, Teng et al. [32] and Zhang et al. [33,34] also considered the influence of steel reinforcement on concrete cover separation. The radial tensile stress distributed in the concrete around steel reinforcement and generated by bond action was applied to simulations. To consider the interaction between concrete cracks and tensile steel reinforcement and FRP plates, Maio et al. [39] implemented a truss model based on an interelement cohesive fracture approach into the simulations of FRP-plated RC beams that failed due to concrete cover separation. Although the calculational accuracy was progressively improved, the calculational procedure and cost was also accordingly increased to limit its practical applications. 

Great efforts on experiments and on analytical and numerical approaches have been made to investigate the concrete cover separation of RC members strengthened by external FRP reinforcement (plates, sheets, or strips), whereas a full understanding of this failure mode is still lacked, and the precise and reliable calculation approach is still in demand. In this paper, a novel stress field-based analytical approach is presented to evaluate the failure strength of concrete cover separation induced at the end of external FRP reinforcement and the corresponding carrying capacity of FRP-strengthened RC beams that failed in concrete cover separation. First, dowel action of steel and FRP reinforcement and the induced concrete splitting, which reflect the interaction between concrete, steel and FRP reinforcement and are commonly ignored in existing investigations, are considered to be the critical factors to result in concrete cover separation and properly incorporated into the establishment of fundamental geometrical relationships of stress field for cracked concrete block. Then the influential factors such as transverse strain and splitting crack of concrete interacted with surrounding steel and FRP reinforcement are comprehensively taken account for the calculation of effective compressive strength of concrete in stress field. Accordingly, the cracking angle to identify the stress field, and the innovative failure criterion are derived to predict the occurrence of concrete cover separation and the related mixed modes of debonding failure; the arrangement details of steel and FRP reinforcement and the cracking status of surrounding concrete and its softening effect are properly considered in the derivation. Subsequently, the indispensable shear component in the effective tensile strain of FRP is taken into account by a simplified stress field approach to evaluate the carrying capacity of strengthened RC beams that failed in end concrete cover separation. Finally, the proposed calculational model is effectively validated by the experimental results available in the literature. 

## 2. Analysis of Mechanical State and Failure Mechanism of Concrete Cover Separation

Before the establishment of analytical model, it is fundamental to understand the mechanical state and failure mechanism of separation of concrete cover at FRP reinforcement end. According to many experimental investigations [4,6,7,9,10,12,13,16,17,18,21,29,32,33,34,46,47], the crack configurations of concrete cover separation (CCS) and interfacial debonding (ID) are schematically shown in Figure 3a,b. In contrast to interfacial debonding that occurs at the interface between FRP and concrete, as shown in Figure 3b, concrete cover separation initiates from an inclined separation crack at the end of FRP reinforcement and then progressively propagates towards the horizontal direction up to the level of the internal tensile steel reinforcement, as shown in Figure 3a. 

More specifically, if the applied shear stress exceeds the bond strength between FRP and concrete, the crack creates at the FRP/concrete interface, which is known as interfacial debonding [4,6,7,21]. Otherwise, if the tensile stress of concrete on inclined plane resulted from the applied shear stress and the possible peeling stress surpasses the effective tensile strength of concrete, the inclined separation crack which is the typical feature of concrete cover separation would be generated [4,6,7,17,21,26,27,29,34,46,47]. Another critical feature is the horizontal crack along the level of internal tensile steel reinforcement, which is not completely contributed by the applied shear stress of FRP reinforcement [32,33,34,46]. As the available experimental investigations [25,30,46] demonstrated that the partial horizontal cracks, constituted of splitting cracks, as shown in Figure 3a, could appear before the formation of inclined crack and were mainly caused by the interaction between concrete and steel and FRP reinforcement. 

If FRP reinforcement is not attached, the horizontal splitting cracks can be also generated because the bond action between steel reinforcement and concrete [48,49,50] and the dowel action of steel reinforcement can result in the non-ignorable tensile stress in the surrounding concrete around steel reinforcement [51,52,53,54,55,56], which are respectively illustrated in Figure 4 and Figure 5. Furthermore, the locations where splitting cracks generally are created [53,56,57] for steel RC beams, as schematically shown in Figure 6 matches the failure modes of concrete cover separation occurring at the maximum moment region and at the end of FRP reinforcement for strengthened RC beams. The coincidence in failure locations indicates that the bond action and dowel action of steel and FRP reinforcement and the induced concrete splitting are closely related to concrete cover separation; it should be considered to be the influential factors to cause concrete cover separation in addition to the shear stress applied on FRP reinforcement, which could be validated by the experimental investigation performed by Al-Saawani et al. [25].

This study mainly focuses on the end concrete cover separation of FRP-strengthened RC beams. Thus, the dominant dowel action of steel and FRP reinforcement and the induced concrete splitting would be considered, and the failure mechanism of concrete cover separation generated at the maximum moment region and the relevant bond action of steel reinforcement is not further discussed.

## 3. Analytical Model to Predict Concrete Cover Separation

Concrete cover between internal steel reinforcement and external FRP reinforcement is divided into discrete blocks by inclined separation cracks along the strengthened RC beam. Based on the conventional assumptions and simplifications, the stress field approach schematically shown in Figure 7 can properly reflect the interaction between concrete, steel and FRP reinforcement and consider the cracking status of surrounding concrete and its softening effect; it is used to assess the mechanical state of cracked concrete block. Meanwhile, the novel failure criterion and failure strength, incorporating the aforementioned influential factors and considering the arrangement details of internal steel reinforcement, are derived to predict concrete cover separation.

### 3.1. Assumptions and Simplifications 

To derive the failure criterion and failure strength of end concrete cover separation, only the mechanical state of cracked concrete block at the end of FRP reinforcement needs to be appraised. Consequently, the fundamental hypotheses about mechanical and geometrical conditions are made to satisfy the requirements of stress field theory for concrete [58,59,60,61,62,63,64,65], which are described as follows and schematically shown in Figure 8.

Two adjacent inclined separation cracks are defined as the boundaries of concrete block. The tensile stress along the inclined cracks reaches the effective tensile strength of concrete, and the shear stress along the inclined cracks is ignored [49,50,54].

The first inclined separation crack straightly extends from the end of FRP reinforcement to the gravity center of the (outer) steel reinforcement; without considering crack spacing, the adjacent inclined and straight separation crack starts at the end of projection of the first inclined crack (the straight line between points A and A’, as shown in Figure 8, is vertical to the bottom and top surfaces); the extension lines of the two inclined cracks connect at one point of intersection [59,60,61,63,64]; and the horizontal upside and underside of concrete block coincide with the gravity center of the steel reinforcement and the FRP/concrete interface, respectively.

The initial splitting crack is horizontally developed along the level of steel reinforcement from the joint point connecting the second inclined crack and the gravity center of the (outer) steel reinforcement to the first inclined crack. The propagation length depends on the splitting degree and is assumed to be larger than the length of the upside of concrete block in this study.

The dowel action is simplified as uniform load distributed on surrounding concrete block [49,54,55,56].

Considering the plasticity and stress redistribution of concrete, the assumption and simplification of uniformly distributed tensile stress along inclined cracks and dowel action would generate the conservatively safe evaluation of failure strength of concrete cover separation, which is specifically validated and discussed in Section 5.

### 3.2. Specifications of the Mechanical and Geometrical Conditions of Stress Field for Cracked Concrete Block 

To establish the geometrical relationships of stress field for cracked concrete block, the mechanical state should be identified. First, the uniformly distributed tensile stress along the inclined separation cracks is defined as $\eta_{ct}f_{ct}$, where the coefficient $\eta_{ct}$ considers the concrete brittleness in tension and is assigned with 0.8 [49,50,54,55,56].

Subsequently, it should be pointed out that the dowel actions of both steel and FRP reinforcement contribute to concrete cover separation according to experimental investigations [10]. However, there is quite limited investigations and calculational approaches about dowel action of FRP reinforcement (sheets, plates, or strips) due to its low rigidity compared to that of steel reinforcement. To facilitate the calculation and analysis of dowel action of steel reinforcement, Fernández Ruiz et al. [49,54,55] and Cavagnis et al. [56] simplified the induced tensile stress in surrounding concrete as the uniform load $f_{c,ef}$ with an upper bound of $\eta_{ct}f_{ct}$ distributed in a certain effective region around steel reinforcement (Figure 9) and then correlated its magnitude with the strain level of steel reinforcement. Consequently, the simplified dowel action can be estimated by Equations (6)–(8) [49,54,55,56]:(6)$$V_{dow}=n_{b}f_{c,ef}b_{ef}l_{ef}$$
(7)$$b_{ef}=min\left(b_{w}/n_{b}-\emptyset_{s},4c_{n}\right)$$
(8)$$l_{ef}=2\emptyset_{s}$$
where $n_{b}$ is the number of the outer steel reinforcement; $b_{ef}$ and $l_{ef}$ are the effective width and length of the distribution region of tensile stress, respectively; and $\emptyset_{s}$ is the diameter of steel reinforcement. The calculational formulation simplified the complex interaction between steel and surrounding concrete, and disclosed the influence of arrangement details and geometries of steel reinforcement on dowel action. 

In this study, the distributed tensile stress on top surface of concrete block is ignored due to the development of splitting crack, and the dowel action $V_{dow}$ of steel reinforcement is assumed to be completely undertaken by the external FRP reinforcement and surrounding concrete; the magnitude of dowel action $V_{dow}$ is approximately estimated by Equations (6)–(8) using the upper bound of $\eta_{ct}f_{ct}$; and the beneficial effect of steel transverse reinforcement on dowel action is conservatively ignored.

Moreover, there exists shear stress acted on the top surface of concrete block surrounded by cracked concrete and steel bars, as schematically shown in Figure 10, and on the bottom surface of concrete block, covered by FRP reinforcement. At the top surface, the shear stress, generated from the complex interaction between cracked concrete and steel bars [66], needs not to be specified. At the bottom surface, the shear stress $\tau_{c}$ and the possible normal tensile stress $\sigma_{ct}$, caused by tensile FRP reinforcement and shown in Figure 8, is further analyzed in the following section. 

The stress state of concrete block concerned in calculations is schematically in Figure 8. One of the geometrical relationships of stress field for cracked concrete block can be established according to the vertical equilibrium of the applied forces, which is expressed by Equation (9):(9)$$\cos\theta_{cr}\eta_{ct}f_{ct}ab_{w}-\sigma_{ctm}ab_{w}=\cos\beta \eta_{ct}f_{ct}bb_{w}+n_{b}f_{c,ef}b_{ef}l_{ef}\;$$
where $\theta_{cr}$ is the angle of the first inclined separation crack, and defined as cracking angle; β is the angle of the second inclined separation crack; a and b are the lengths of underside and upside of concrete block, respectively; and $\sigma_{ctm}$ is the assumed possible average normal tensile stress.

Replacing $f_{c,ef}$ in Equation (9) with $\eta_{ct}f_{ct}$ and rearranging the equation, the geometrical relationship can be simplified as Equation (10):(10)$$a\cos\theta_{cr}-\frac{\sigma_{ctm}a}{\eta_{ct}f_{ct}}=b\cos\beta +\frac{n_{b}b_{ef}l_{ef}}{b_{w}}$$

Defining the parameters μ and λ as:(11)$$\mu =\frac{\sigma_{ctm}}{\eta_{ct}f_{ct}}$$
(12)$$\lambda =\frac{n_{b}b_{ef}l_{ef}}{b_{w}}$$

Accordingly, Equation (10) can be further simplified into Equation (13) explicitly representing the relationship between cracking angle $\theta_{cr}$ and inclined angle *β*.
(13)$$\frac{a\left(\cos\theta_{cr}-\mu \right)-\lambda }{b}=\cos\beta$$

In addition, the relationship between cracking angle $\theta_{cr}$ and inclined angle β can be established according to geometrical relationship and expressed by Equation (14):(14)$$\frac{c_{b}}{b}=\frac{a\tan\theta_{cr}}{b}=\tan\beta$$
where $c_{b}$ is the thickness of concrete block, and measured from the bottom surface of concrete beam to the gravity center of the (outer) tensile steel reinforcement.

A parameter ξ is introduced to express the ratio of lengths of underside to upside of concrete block, which is shown by Equation (15):(15)$$\xi =\frac{a}{b}$$

Consequently, through combining Equations (13) and (14), the relationship between geometrical parameter ξ and cracking angle $\theta_{cr}$ can be obtained and expressed by Equation (16):(16)$$\xi^{2}\left(1+\xi^{2}\tan^{2}\theta_{cr}\right)={\left(\cos\theta_{cr}-\mu -\lambda /c_{b}\tan\theta_{cr}\right)}^{-2}$$ Solving Equation (16), the geometrical parameter ξ can be explicitly expressed by cracking angle $\theta_{cr}$, and the expressions are shown by Equations (17) and (18):(17)$$\xi =\sqrt{\left(\frac{\sqrt{1+4\hbar \tan^{2}\theta_{cr}}-1}{2c_{b}}\right)}$$
(18)$$\hbar ={\left(\cos\theta_{cr}-\mu -\lambda /c_{b}\tan\theta_{cr}\right)}^{-2}$$
where ℏ is a parameter.

With the geometrical parameter ξ, the relationship represented by Equation (19) between the principal compressive stresses at the center and at the bottom surface of the micro concrete strut closely adjacent to the first inclined crack whose extension line goes through the point O of intersection (Figure 8) can be derived according to the stress field theory for concrete [59,60,61,63,64].
(19)$$f_{cp}=\frac{f_{cp0}}{2}\left(1+1/\xi \right)$$
where $f_{cp}$ is the local principal compressive stress at the bottom surface adjacent to external FRP reinforcement; and $f_{cp0}$ is the local principal compressive stress at the center of the micro concrete strut. 

Furthermore, based on the geometrical relationship demonstrated in Figure 8, the average angle $\theta_{m}$ of stress field for cracked concrete block can be also expressed by the geometrical parameter ξ and cracking angle $\theta_{cr}$, and shown by Equation (20): (20)$$\tan2\theta_{m}=\tan\left(\theta_{cr}+\beta \right)=\frac{\left(1+\xi \right)\tan\theta_{cr}}{1-\xi \tan^{2}\theta_{cr}}$$ The derived geometrical relationships indicate that the cracking angle $\theta_{cr}$ is a fundamental parameter to identify the stress field for cracked concrete block.

### 3.3. Failure Criterion 

#### 3.3.1. Critical Mechanical State of Concrete Cover Separation

One of the critical failure conditions of concrete cover separation is concrete cracking along the inclined separation cracks at the end of FRP reinforcement. Thus, the transverse stress of micro concrete struts (Figure 8) adjacent to inclined crack reaches effective tensile strength of concrete $\eta_{ct}f_{ct}$. Correspondingly, the transverse tensile strain or called principal tensile strain $\varepsilon_{c,1}$ of micro concrete strut is assumed as $\varepsilon_{ct}$ which is the cracking strain of tensile concrete and can be estimated by Equation (21) [67,68]:(21)$$\varepsilon_{ct}=\frac{f_{ct}}{E_{D}}=\frac{f_{ct}}{0.83E_{c}}\;$$
where $E_{D}$ is the dynamic Young’s modulus of concrete; and $E_{c}$ is the static Young’s modulus of concrete.

Relationship between the principal compressive strain $\varepsilon_{c,2}$ and the principal tensile strain $\varepsilon_{c,1}$ of micro concrete strut can be established according to the Poisson’s ratio ν and represented by Equation (22) [68,69]:(22)$$\varepsilon_{c,2}=\varepsilon_{c,1}/\nu =\varepsilon_{ct}/\nu \;$$

Please note that the principal compressive strain $\varepsilon_{c,2}$ refers to the longitudinal strain in the center of micro concrete strut, as shown in Figure 8; furthermore, the Poisson’s ratio ν is assigned with 0.2 for cracking concrete [68,69]. The corresponding principal compressive stress $f_{cp0}$ in the center of micro concrete strut can be assessed using the Hognestad parabola represented by Equation (23) [70]:(23)$$f_{cp0}=\left(\frac{\varepsilon_{c,2}^{2}+2\varepsilon_{c,2}\varepsilon_{c0}}{\varepsilon_{c0}^{2}}\right)f_{ce}$$
where $\varepsilon_{c0}$ is the concrete strain corresponding to the peak stress in the stress-strain constitutive curve of compressive concrete; and $f_{ce}$ is the effective compressive strength of concrete.

#### 3.3.2. Effective Compressive Strength of Concrete

The effective compressive strength $f_{ce}$ of concrete in stress field is different from that in uniaxial compressive state and is generally lower than the cylinder compressive strength $f_{c}’$ of concrete [58,59,60,61,62,63,64,65]. Plenty of factors affect the compressive strength of concrete in stress field; plasticity of concrete, transverse strain of stress field of concrete, and crack width of cracked concrete, for example [49,50,54,55,56,58,59,60,61,62,63,64,65]. Currently, the effective compressive strength of concrete can be estimated by modifying the cylinder compressive strength $f_{c}’$ of concrete with a series of reducing coefficients reflecting the aforementioned softening effect.

Consequently, the effective compressive strength of concrete in stress field is calculated by Equation (24):(24)$$f_{ce}=\eta_{fc}\eta_{\varepsilon }\eta_{w}f_{c}’\;$$
where $\eta_{fc}$ is the plasticity of concrete coefficient [60]; $\eta_{\varepsilon }$ is the transverse strain influential coefficient [58,59,60,61,62,63,64,65]; and $\eta_{w}$ is the cracked concrete influential coefficient [71,72,73,74,75,76,77,78,79]. The brief illustration of the reducing coefficients is given in Appendix A.

Equation (24) indicated that the generation and development of splitting crack would significantly reduce the effective compressive strength of concrete and affect the failure strength of concrete cover separation discussed as follows. 

#### 3.3.3. Failure Strength of Concrete Cover Separation and Cracking Angle of Concrete Block

To establish the failure criterion of concrete cover separation, stress state of a local part of micro concrete strut adjacent to the first inclined separation crack and external FRP reinforcement is shown in Figure 8b. The effective tensile stress $\eta_{ct}f_{ct}$ along concrete crack is considered to be the principal tensile stress of concrete strut. The corresponding principal compressive stress $f_{cp}$ is vertical to the effective tensile stress and correlated with the principal compressive stress $f_{cp0}$ at the center of concrete strut by Equation (19). In addition, there exists local shear stress $\tau_{c}$ and the possible tensile stress $\sigma_{ct}$ due to peeling force applied on the bottom surface of micro concrete strut stemming from external FRP reinforcement. The conservatively safe case of normal compressive stress applied at the bottom of concrete strut is not studied herein.

The principal tensile stress $\eta_{ct}f_{ct}$ and principal compressive stress $f_{cp}$ identify a Mohr’s circle of stress with a radius of $\left(f_{cp}+\eta_{ct}f_{ct}\right)/2$ and a center of $\left(f_{cp}-\eta_{ct}f_{ct}\right)/2$ in the normal stress σ -shear stress τ coordinate system, which is shown in Figure 11. Please note that the positive axis of abscissa represents the normal compressive stress in this study. As discussed before, the principal compressive stress $f_{cp}$ is not completely characterized by material properties. For the identical material properties, the characterization of a Mohr’s circle of stress would vary with the development of spitting crack width, which can be schematically illustrated by the Mohr’s circles respectively characterized by the principal compressive stresses $f_{cp}$ and (the softened) $f{’}_{cp}$ shown in Figure 11a. Consequently, the following analysis of failure state of micro concrete strut is divided into two cases according to the comparison between principal tensile stress $\eta_{ct}f_{ct}$ and principal compressive stress $f_{cp}$.

Case 1: $f_{cp}\geq \eta_{ct}f_{ct}$

Generally, the principal compressive stress $f_{cp}$ of concrete strut is larger than its principal tensile stress $\eta_{ct}f_{ct}$. The corresponding Mohr’s circle of stress is schematically shown in Figure 11a. It can be seen that the shear stress $\tau_{c}$ increases as the corresponding tensile stress $\sigma_{ct}$ decreases; as the tensile stress $\sigma_{ct}$ equals to zero, the shear stress $\tau_{c}$ reaches the maximum $\tau_{cmax}$, which is also the maximum local shear stress along the bottom surface according to the stress field theory for concrete [63,64]. Accordingly, the maximum local shear stress $\tau_{cmax}$ of concrete strut is derived and represented by Equation (25):(25)$$\tau_{cmax}=\sqrt{\eta_{ct}f_{ct}f_{cp}}\;$$

In contrast to most formulations of debonding failure strength, characterized by the individual material properties and geometries of concrete and external FRP reinforcement, the derived strength model additionally incorporates the influential factors such as interaction between concrete, steel and FRP reinforcement, cracking status of surrounding concrete and its softening effect, and arrangement details of internal steel reinforcement and external FRP reinforcement. Moreover, as the Mohr’s circle for stress of micro concrete in Figure 11 and Equation (25) shown that the shear strength $\tau_{cmax}$ would be improved by imposing the moderate normal compressive stress and restricting the formation and development of crack. Therefore, various anchorage devices [80,81] are widely used in practice to prevent concrete cover separation.

On the other hand, to ensure the occurrence of concrete cover separation, the maximum local shear stress $\tau_{cmax}$ should satisfy the following requirement about bond strength $\tau_{fu}$ of external FRP reinforcement [82,83], which is expressed by Equation (26):(26)$$\tau_{fmax}=\tau_{cmax}\frac{b_{w}}{b_{f}}\leq \tau_{fu}=\frac{\sqrt{f_{ct}f_{c}’}}{2}\;$$
where $\tau_{fmax}$ is the maximum bond stress between FRP reinforcement and concrete at concrete cover separation; and $b_{f}$ is the width of FRP reinforcement externally bonded to concrete surface. As the maximum bond stress $\tau_{fmax}$ equals to the bond strength $\tau_{fu}$ in Equation (26), the mixed mode of interfacial debonding and cover separation is present, which is also commonly observed in experiments [9,10,16,47] and schematically shown in Figure 12a. It is worth pointing out that the bond strength $\sqrt{f_{ct}f_{c}’}/2$ between FRP reinforcement and concrete [82,83] in Equation (26) can be replaced by the more detailed and accurate failure criterion for other composite materials to meet requirement of analysis. 

Another possible mixed mode of cover separation and sliding failure is schematically shown in Figure 12b and is seldom observed in experiments. It occurs as the Mohr’s circle of stress touches the Mohr-Coulomb failure criterion [64,82,83] for the principal compressive stress $f_{cp}$ being pronouncedly larger than principal tensile stress $\eta_{ct}f_{ct}$, as shown in Figure 13. According to the stress conditions at the critical state, namely the Mohr’s circle of stress with red color in Figure 13, the maximum principal compressive stress $f_{cp,max}$ can be computed by Equation (27): (27)$$f_{cp,max}=\eta_{fc}\eta_{w}f_{c}^{’}-\frac{1+\sin\varphi_{f}}{1-\sin\varphi_{f}}\eta_{ct}f_{ct}\;\;$$
where $\varphi_{f}$ is the internal angle of friction of concrete and assigned with 37.2° [64,71]. Hence, the principal compressive stress $f_{cp}$ should also satisfy the condition of $f_{cp}\leq f_{cp,max}$ to guarantee the occurrence of concrete cover separation.

In addition, according to the geometrical relationships of Mohr’s circle of stress shown in Figure 11a, the cracking angle $\theta_{cr}$ can be calculated using Equation (28) and, meanwhile, should satisfy the requirement of $0<\theta_{cr}\leq 45{}^{\circ}$.
(28)$$\cos2\theta_{cr}=\frac{f_{cp}-\eta_{ct}f_{ct}}{f_{cp}+\eta_{ct}f_{ct}}=\frac{1-\frac{\eta_{ct}f_{ct}}{f_{cp}}}{1+\frac{\eta_{ct}f_{ct}}{f_{cp}}}$$

Based on Equation (28) and Figure 11a, it can be further inferred that the cracking angle $\theta_{cr}$ increases as the principal compressive stress $f_{cp}$ reduces, which agrees with relevant conclusion of the stress field theory for reinforced concrete [58,59,60,61,62,63,64,65].

Case 2: $f_{cp}<\eta_{ct}f_{ct}$

By contrast, Case 2 does commonly occur due to the extremely low principal compressive stress of $f_{cp}$. The corresponding Mohr’s circle of stress is schematically shown in Figure 11b. It can be seen that as the tensile stress $\sigma_{ct}$ decreases the local shear stress $\tau_{c}$ increases to the maximum and then decreases. The maximum shear stress $\tau_{cmax}$ corresponds the center of Mohr’s circle of stress locating at the negative abscissa. It means that the shear stress $\tau_{c}$ reaches the maximum shear stress of $\tau_{cmax}=\left(f_{cp}+\eta_{ct}f_{ct}\right)/2$ when the tensile stress $\sigma_{ct}=\left(\eta_{ct}f_{ct}-f_{cp}\right)/2$ is applied on bottom surface of concrete strut. Furthermore, the corresponding cracking angle $\theta_{cr}$ is 45° as shown in Figure 11b.

So far, the mechanical state of micro concrete strut adjacent to the first inclined separation crack shown in Figure 8 can be identified. Furthermore, it can be found that the cracking angle $\theta_{cr}$ identifies the stress field for global cracked concrete block and relates to the failure strength of concrete cover separation. Subsequently, the detailed steps to estimate cracking angle $\theta_{cr}$ are introduced as follows.

Select a cracking angle $\theta_{cr}$
($\theta_{cr}$ < 45°);Specify the geometrical parameter ξ, cylinder compressive strength $f_{c}’$, tensile strength $f_{ct}$, and static Young’s modulus of concrete $E_{c}$;Calculate the cracking strain of tensile concrete $\varepsilon_{ct}$ and the principal compressive strain $\varepsilon_{c,2}$;Select a splitting cracking width of w and compute the cracked concrete influential coefficient $\eta_{w}$;Figure out the plasticity of concrete coefficient $\eta_{fc}$, transverse strain influential coefficient $\eta_{\varepsilon }$, effective compressive strength of concrete $f_{ce}$, and principal compressive stresses $f_{cp0}$ and $f_{cp}$;Check the correctness of principal compressive stress $f_{cp}$ according to the maximum local shear stress $\tau_{cmax}$, estimated by Equations (25) and (26), and the maximum principal compressive stress $f_{cp,max}$, derived by Equation (27);If Step 6 is false, adjust the value of w  and then repeat Steps 4–6;If Step 6 is true, calculate the cracking angle $\theta_{cr}$ from Equation (28) using the obtained material properties and relevant parameters;If the computed cracking angle $\theta_{cr}$ in step 8 is larger than 45°, calculate the cracking angle $\theta_{cr}$ and maximum shear stress $\tau_{cmax}$ defined in Case 2;Check the computed cracking angle $\theta_{cr}$ in step 8 with the assumed one in Step 1;

If Step 10 is false, repeat steps 1–8 and 10; and

If Step 10 is true, obtain the desired cracking angle $\theta_{cr}$.

Please note that under the condition of satisfying the requirements of failure strength listed in step 6 and the lower bound of splitting crack width of 0.015 $\emptyset_{s}$, which is illustrated in Appendix A, the splitting cracking width w with the initial value of 0.015 $\emptyset_{s}$ should be as little as possible to estimate the lower bound of cracking angle.

Consequently, an analysis flowchart to estimate cracking angle $\theta_{cr}$ is illustrated in Figure 14.

#### 3.3.4. Effective Tensile Strain of FRP Reinforcement Corresponding to Concrete Cover Separation

As the cracking angle $\theta_{cr}$ is known, the average cracking angle $\theta_{m}$ of the stress field for cracked concrete block can be computed by Equation (20). Then the corresponding simplified average stress field for cracked concrete block with a shape of right triangle subject to biaxial tension-compression load is established and shown in Figure 15. Specifically, the average principal compressive stress $f_{cpm}$ is applied on the leg corresponding to the average cracking angle $\theta_{m}$, and the tensile stress $\eta_{ct}f_{ct}$ is applied on the other leg. The average shear stress $\tau_{cm}$ and the aforementioned possible average tensile stress $\sigma_{ctm}$, are uniformly distributed on the hypotenuse. It should be clarified that the simplified average stress field, characterized by the average mechanical state of the global cracked concrete block, is not identical to that of local micro concrete strut shown in Figure 8b. 

Referring to the aforementioned analytical approaches of local shear stress, the average shear stress $\tau_{cm}$ can be simply represented by the tensile stress $\eta_{ct}f_{ct}$ and the average cracking angle $\theta_{m}$ without considering the average principal compressive stress $f_{cpm}$. Specifically, for Case 1 without the average tensile stress $\sigma_{ctm}$, the corresponding Mohr’s circle of stress is shown in Figure 16a. Accordingly, the average shear stress $\tau_{cm}$ can be evaluated by Equation (29).
(29)$$\tau_{cm}=\frac{\eta_{ct}f_{ct}}{\tan\theta_{m}}$$

For the scarcely occurring Case 2 with the average tensile stress $\sigma_{ctm}$, the average cracking angle $\theta_{m}$ is larger than 45° and the corresponding Mohr’s circle of stress is schematically shown in Figure 16b. Equation (29) can be also used to estimate the upper bound of the average shear stress $\tau_{cm}^{u}$.

With the estimated average shear stress $\tau_{cm}$ or $\tau_{cm}^{u}$, the resultant force $T_{f}$ of FRP reinforcement to result in concrete cover separation can be computed by Equation (30):(30)$$T_{f}=\frac{\eta_{ct}f_{ct}}{\tan\theta_{m}}b_{w}c_{b}\cot\theta_{cr}$$

The effective tensile strain of FRP corresponding to the occurrence of concrete cover separation is evaluated by Equation (31):(31)$$\varepsilon {’}_{fe}=\frac{\eta_{ct}f_{ct}}{\tan\theta_{m}E_{f}A_{f}}b_{w}c_{b}\cot\theta_{cr}$$

It should be pointed out that the effective tensile strain of FRP corresponding to concrete cover separation is not identical to the one at the maximum bending moment section that failed in flexure, which is assumed to be sufficient and not the critical factor in this study. 

## 4. Analytical Model of Carrying Capacity of the FRP-Strengthened RC Beams That Failed in Concrete Cover Separation

The location of end concrete cover separation renders the huge difficulties in precise evaluation of carrying capacity of RC beams strengthened with FRP reinforcement. Hence, most studies merely paid attention to assessing the effective tensile strain of FRP corresponding to concrete cover separation [26,27,28,29,30,31,38]. Only limited studies presented numerical and analytical approaches to estimate the carrying capacity of RC beams that failed in concrete cover separation [33,34,36]. Although the available numerical simulations can accurately predict the ultimate state of strengthened RC beams, the sophisticated modeling techniques and lengthy calculation process limit the practical design and analysis. On the other hand, the analytical approaches could highly reduce the calculation efforts and time-costs. The excessive simplifications in analysis usually result in dissatisfactory evaluation results, particularly for the ignorance of pronounced shear deformation [33,34,35,36,84]. In this section, an analytical approach, able to comprehensively consider the influence of flexural–shear action on tensile strain of FRP reinforcement, is proposed to predict the carrying capacity of FRP-strengthened RC beams that failed in concrete cover separation.

### 4.1. Background to the Proposed Model

For the intermediate crack-induced concrete cover separation, as shown in Figure 1b, an extremely large proportion of effective tensile strain of FRP reinforcement is exhausted by the flexural action of strengthened RC beams [57,84]. Hence, the conventional plane-section analysis of flexural response can be performed in a simple manner to obtain the satisfactory estimation of carrying capacity corresponding to the effective tensile strain of FRP. However, this analytical approach is not suited for the strengthened RC beams that failed in end concrete cover separation, since the shear action would significantly take up the effective tensile strain of FRP [84]. To consider the influence of shear action on carrying capacity of the FRP-strengthened concrete beams and facilitate the calculation, the uniaxial shear-flexural model (USFM) [85] can be extended and used. The fundamental strategy of USFM is that the flexural response and shear response of a RC beam can be estimated by the combination of plane-section analysis and stress field theory. Based on the fundamental strategy and considering the specific retrofitting configurations, a simplified analytical approach for predicting the carrying capacity of FRP-strengthened RC beams that failed in end concrete cover separation is presented and introduced.

### 4.2. Analytical Model

By contrast with the conventional sectional analysis performed at the critical section with the maximum bending moment, the presented analytical approach focuses on the section where the concrete cover is completely separated, namely the end of the first inclined separation crack (Figure 17). On the studied section, the tensile strain $\varepsilon_{f}$ of FRP reinforcement is considered to be the sum of flexural strain $\varepsilon_{f,f}$ and shear strain $\varepsilon_{f,s}$, and expressed by Equation (32) [62,84]:(32)$$\varepsilon_{f}=\varepsilon_{f,f}+\varepsilon_{f,s}$$

Subsequently, the analyses of flexural and shear behavior of strengthened RC beams are respectively performed to specify the flexural strain $\varepsilon_{f,f}$ and shear strain $\varepsilon_{f,s}$.

#### 4.2.1. Flexural Behavior

Flexural strain $\varepsilon_{f,f}$ can be assessed through the conventional plane-section analysis based on equilibrium and compatibility principles, as shown in Figure 18. The resultant force T at tensile region is expressed by Equations (33) and (34):(33)$$T=A_{s}\sigma_{s,f}+A_{f}\sigma_{f,f}\;$$
(34)$$\sigma_{s,f}=E_{s}\varepsilon_{s,f}\leq f_{sy},\;\sigma_{f,f}=E_{f}\varepsilon_{f,f}\leq f_{fu}$$
where $A_{s}$ is the area of tensile steel longitudinal reinforcement; $\sigma_{s,f}$ and $\sigma_{f,f}$ are the tensile stresses of steel reinforcement and FRP reinforcement due to flexural behavior, respectively; $\varepsilon_{s,f}$ and $\varepsilon_{f,f}$ are the tensile steel strain and FRP strain due to flexural behavior, respectively; $f_{sy}$ and $f_{fu}$ are the yield strength of steel longitudinal reinforcement and the tensile strength of FRP reinforcement, respectively. The tensile behavior of steel longitudinal and transverse reinforcement follows a bilinear stress-strain constitutive law, and the strain hardening is ignored.

The resultant force C at compressive region is expressed by Equations (35) and (36):(35)$$C=C_{c}+A{’}_{s}\sigma {’}_{s,f}\;$$
(36)$$\sigma {’}_{s,f}=E{’}_{s}\varepsilon {’}_{s,f}\leq f{’}_{sy}$$
where $\sigma {’}_{s,f}$ and $\varepsilon {’}_{s,f}$ are the compressive stress and strain of steel due to flexural behavior, respectively; $A{’}_{s}$, $E{’}_{s}$, and $f{’}_{sy}$ are the area, elastic modulus, and yield strength of compressive steel longitudinal reinforcement, respectively; and $C_{c}$ is the concrete resultant force at compressive region and can be predicted by the Hognastad’s curve of concrete [70], represented by Equations (37)–(39):(37)$$C_{c}=\alpha_{1}f_{c}^{’}\beta_{1}cb_{w}\;$$
(38)$$\alpha_{1}\beta_{1}=\frac{\varepsilon_{c}}{\varepsilon_{c0}}-\frac{1}{3}{\left(\frac{\varepsilon_{c}}{\varepsilon_{c0}}\right)}^{2}\;$$
(39)$$\beta_{1}=\frac{4-\varepsilon_{c}/\varepsilon_{c0}}{6-2\varepsilon_{c}/\varepsilon_{c0}}\;$$
where $\alpha_{1}$ and $\beta_{1}$ are the parameters of the equivalent stress block of compressive concrete; c is the depth of compressive region; and $\varepsilon_{c}$ is the strain of concrete extreme compression fiber.

According to the common assumption that the plane section remains plane, the depth of compressive region c and the strain of concrete extreme compression fiber $\varepsilon_{c}$ can be computed by Equations (40) and (41), respectively:(40)$$c=d_{f}-\varepsilon_{f,f}/\varphi$$
(41)$$\varepsilon_{c}=\varphi c\;$$
where φ is the curvature of beam section; and $d_{f}$ is the depth of FRP reinforcement and is the sum of depth of beam section h and half of thickness of FRP reinforcement $t_{f}$, which can be ignored in calculation.

Similarly, the compressive strain $\varepsilon {’}_{s,f}$ and tensile strain $\varepsilon_{s,f}$ of steel longitudinal reinforcement due to flexural behavior are expressed by Equations (42) and (43), respectively:(42)$$\varepsilon {’}_{s,f}=\varphi \left(c-a{’}_{s}\right)$$
(43)$$\varepsilon_{s,f}=\varphi \left(d_{s}-c\right)$$
where $a{’}_{s}$ and $d_{s}$ are the depths of compressive and tensile steel reinforcement, respectively. 

Consequently, the bending moment $M\left(\varepsilon_{f,f}\right)$ applied on the studied beam section corresponding to flexural strain $\varepsilon_{f,f}$ of FRP longitudinal reinforcement can be computed by Equation (44):(44)$$M\left(\varepsilon_{f,f}\right)=A{’}_{s}\sigma {’}_{s,f}\left(c-a{’}_{s}\right)+\alpha_{1}\beta_{1}f_{c}^{’}cb_{w}\left(c-\frac{\beta_{1}}{2}c\right)+A_{s}\sigma_{s,f}\left(d_{s}-c\right)+A_{f}\sigma_{f,f}\left(d_{f}-c\right)$$

#### 4.2.2. Shear Behavior

The shear force V is considered constantly distributed along the span l for simply supported beams and is computed by Equation (45):(45)$$V=M\left(\varepsilon_{f,f}\right)/\left(l_{0}+c_{b}\cot\theta_{cr}\right)\;$$
where $l_{0}$ is the distance from end of FRP reinforcement to the nearest support, as illustrated in Figure 17.

Considering the combination of steel and FRP longitudinal reinforcement, the calculational formulation based on the modified compression field theory (MCFT) [58,62] and the cracked membrane model (CMM) [61] is extended to estimate the strain of $\varepsilon_{f,s}$ contributed by shear behavior and represented by Equation (46):(46)$$\varepsilon_{f,s}=\frac{V\cot\vartheta }{2\left(E_{s}A_{s}+E_{f}A_{f}\right)}\;$$
where ϑ is the inclination angle of the stress field in shear zone of a strengthened RC beam, and different from the cracking angle of concrete block $\theta_{cr}$, as schematically shown in Figure 17.

The development of angle of ϑ can be estimated according to a series of conditions about equilibrium and compatibility based on the MCFT [58,62] and the CMM [61], whereas the lengthy iterative calculation would significantly increase the difficulties in assessment of carrying capacity of strengthened RC beams. On the other hand, Aprile et al. [17,18] reported that the inclination angle of ϑ tends to be stable at concrete cover separation through experimental investigations. The calculational method, based on compression field theory (CFT) [86], was modified by Aprile et al. [17,18] to estimate the lower bound of inclination angle $\vartheta_{u}$ and expressed by Equations (47)–(49):(47)$$\vartheta_{u}=arc\tan\sqrt[{4}]{\frac{1+\frac{1}{\alpha_{sl}\rho_{sl}+\alpha_{fl}\rho_{fl}}}{1+\frac{1}{\alpha_{sv}\rho_{sv}}}}\;$$
(48)$$\rho_{sl}=\frac{A_{s}}{b_{w}z},\rho_{fl}=\frac{A_{f}}{b_{w}z},\rho_{sv}=\frac{A_{sv}}{b_{w}s_{v}}\;$$
(49)$$\alpha_{sl}=\frac{E_{s}}{E_{c}},\;\alpha_{fl}=\frac{E_{f}}{E_{c}},\alpha_{sv}=\frac{E_{sv}}{E_{c}}\;$$
where $\alpha_{sl}$, $\alpha_{fl}$, and $\alpha_{sv}$ are the steel longitudinal reinforcement, FRP longitudinal reinforcement, and steel transverse reinforcement to concrete homogenization coefficients, respectively; $\rho_{sl}$, $\rho_{fl}$, and $\rho_{sv}$ are the geometrical steel longitudinal reinforcement ratio, FRP longitudinal reinforcement ratio, and steel transverse reinforcement ratio, respectively; $A_{sv}$ is the steel transverse reinforcement area; $s_{v}$ is the spacing of steel transverse reinforcement; $E_{sv}$ is the elastic modulus of steel transverse reinforcement; z is the depth of flexural lever arm of beam section and can be approximately assessed by 0.9d; and d is the effective depth of beam section [17,18,87].

Moreover, based on the solution of concrete plasticity [64], the upper bound of inclination angle $\theta_{l}$ can be assessed by Equations (50) and (51):(50)$$\vartheta_{l}=arc\tan\sqrt{\frac{\psi }{1-\psi }}\leq 45{}^{\circ}\;$$
(51)$$\psi =\rho_{sv}\frac{f_{yv}}{f_{c}’}\;$$
where ψ is the mechanical parameter; and $f_{yv}$ is the yield strength of steel transverse reinforcement.

To facilitate calculation, the inclination angle of ϑ. is assessed herein using the average of lower bound $\vartheta_{l}$ and upper bound $\vartheta_{u}$ of inclination angle and represented by Equation (52):(52)$$\vartheta =\frac{\vartheta_{l}+\vartheta_{u}}{2}\;$$

### 4.3. Analytical Process

Considering crack spacing, the starting point of the second inclined separation crack, where the effective tensile strain of FRP at concrete cover separation is derived according to stress field approach, may be not coincident with the studied beam section, as the points A and A’ illustrated in Figure 7. The effective tensile strain of FRP solved by Equation (31), $\varepsilon {’}_{fe}$, needs to be modified by Equation (53) to perform the aforementioned analysis of carrying capacity of strengthened RC beams.
(53)$$\varepsilon_{fe}=\kappa \varepsilon {’}_{fe};\;\left(\kappa \leq 1.0\right)\;$$

The modification coefficient κ is defined by Equation (54):(54)$$\kappa =\frac{c_{b}\cot\theta_{cr}}{s_{f}}\;$$
where $s_{f}$ is the length of cracked concrete block at the level of FRP reinforcement and can be assessed by Equation (55):(55)$$s_{f}=\zeta s_{rm}\;$$
where ζ is defined as the amplification coefficient of average crack spacing of $s_{rm}$ and is identified in the following section; and $s_{rm}$ can be assessed by Equations (56) and (57) [17,18,87,88,89,90]:(56)$$s_{rm}=\left(2c_{n}+0.25k_{1}k_{2}\frac{\emptyset_{s}}{\rho_{eff}}\right)\;$$
(57)$$\rho_{eff}=\frac{A_{s}+A_{f}E_{f}/E_{s}}{A_{c,eff}}\;$$
where $k_{1}$ is a coefficient considering bond characteristics of steel reinforcement, taken as 0.8 for deformed reinforcement and 1.6 for smooth one; $k_{2}$ is a factor accounting for the distribution of tensile stress within beam section and assigned with 0.5; $\rho_{eff}$ is the effective reinforcement ratio; and $A_{c,eff}$ is the effective tensile area of concrete in flexural member, and is estimated by $2.5b_{w}\left(h-d\right)$ that should be not more than $\left(h-c\right)b_{w}/3$ [17,18,87,88,89,90].

Subsequently, the detailed process to assess the carrying capacity of strengthened RC beams that failed in concrete cover separation is illustrated as follows.

Select a flexural strain $\varepsilon_{f,f}$;Select a curvature φ;Figure out the depth of compressive region c, the strain of concrete extreme compression fiber $\varepsilon_{c}$, and the parameters of the equivalent stress block $\alpha_{1}$ and $\beta_{1}$;Specify the reinforcement stresses of $\sigma_{s,f}$, $\sigma_{f,f}$ and $\sigma {’}_{s,f}$ in flexural behavior;Compute the resultant force T at tensile region and C at compressive region;If T is not equal to C, repeat steps 2–5;If T is equal to C, compute the moment $M\left(\varepsilon_{f,f}\right)$, the shear V, and the strain $\varepsilon_{f,s}$; Compute the strain of FRP reinforcement $\varepsilon_{f}$ under flexural–shear action and the effective tensile strain of FRP reinforcement $\varepsilon_{fe}$;If $\varepsilon_{f}$ is not equal to $\varepsilon_{fe}$, repeat steps 1–8; andIf $\varepsilon_{f}$ is equal to $\varepsilon_{fe}$, obtain the desired carrying capacity of $M\left(\varepsilon_{f,f}\right)$ and V of a strengthened RC beam.

Correspondingly, an analysis flowchart to assess the carrying capacity of strengthened RC beams is illustrated in Figure 19.

## 5. Validations and Discussions

To validate the proposed approach, the strengthened RC beams that failed in concrete cover separation with the desirable data in the available literature [9,14,16,22,23,28] were collected. All the RC beams were reinforced with tensile and compressive steel longitudinal reinforcement, and steel transverse reinforcement. Externally boned FRP reinforcement (sheets or plates) was employed to strengthen the RC beams, whereas the FRP reinforcement was not applied along the full spans. A certain distance between FRP reinforcement end and the nearby support was intentionally set. Moreover, all the collected beam specimens were subject to three-point bending tests or four-point bending tests with static loads and failed in concrete cover separation at the end of FRP reinforcement. The geometrical parameters, material properties, reinforcement arrangements, configurations, and anchorage conditions of the failed shear spans are primarily concerned in this investigation and listed in Table 1 and Table 2. The cracking angles of separated concrete blocks are listed in Table 3.

With the proposed analytical model, the calculation of cracking angles of concrete blocks was first performed. The calculational results ($\theta_{cr,cal}$) and comparisons against the experimental results ($\theta_{cr,exp}$) are listed in Table 3 and shown in Figure 20. The statistical results demonstrate that the mean and standard deviation of the ratio between calculations and experimental results are 0.81 and 0.15, respectively, which show the satisfactory accuracy of the proposed analytical approach. Furthermore, Figure 20 indicates that most predictions fall in the acceptable range of cracking angles. The predicted cracking angle, obtained by the proposed approach, is the lower bound of cracking angle due to the conservative assumption of splitting crack width and the consideration of plasticity and stress redistribution of concrete. Therefore, the mean of the ratio between calculations and experimental results in statistics is less than 1.0; furthermore, each prediction is not larger than its corresponding experimental result. The experimental results successfully verified the present analytical approach. In addition, the results also indicate that considering the presence and development of splitting crack is critical to the accurate prediction of cracking angle and the corresponding failure strength. To further improve the accuracy in evaluation of cracking angle, the width of splitting crack, relevant with the cracked concrete influential coefficient $\eta_{w}$, still needs to be identified; moreover, the mechanical state of cracked concrete, and the mixed mode of end debonding failure should be further investigated.

Subsequently, incorporating the effective tensile strain of FRP reinforcement derived from the calculated cracking angles of concrete blocks, carrying capacities of the strengthened RC beams are assessed. In calculations, the effect of amplification coefficient of average crack spacing ζ on the model accuracy is investigated by assigning with the allowable values of 1.35, 1.30, 1.25, and 1.20, respectively. The calculational results are denoted by $V_{cal}^{\zeta }$ and listed in Table 3. To make further comparisons, the calculational results represented by $V_{cal}^{0}$ without considering the modification of crack spacing on effective strain of FRP reinforcement, namely κ=1.0, are also listed; in addition, using the failure strength assessed by the conventional concrete tooth model, carrying capacities of the strengthened RC beams are calculated and reported ($V_{cal}^{ctm}$) in Table 3. Meanwhile, the calculational results ($V_{cal}$) and experimental results ($V_{exp}$) are compared and shown in Figure 21.

The statistical results in Table 3 indicate that the modification coefficient κ, a function of crack spacing, has a significant effect on calculational accuracy; the mean and standard deviation of the ratio between calculations and experimental results are 0.97 and 0.35, respectively, as the amplification coefficient of average crack spacing ζ is assigned with 1.20, which yields the best predictions. By contrast, the calculational results based on the conventional concrete tooth model is less satisfactory than those conservatively safe predictions obtained by the proposed analytical approach; as the length of concrete block is defined as the commonly used average crack spacing, the mean and standard deviation of the ratio between calculations and experimental results are 1.71 and 1.10, respectively; the concrete tooth model overestimated the failure strength of concrete cover separation. Furthermore, the discrepancies would increase as the length of concrete block increase. The similar evaluation conclusions about concrete tooth model can be found in other references [91,92]. Since the sophisticated experimental investigations are very limited, the further validations of the presented analytical model and comparisons between other well-known models still need to be performed in the future.

## 6. Conclusions

A novel analytical approach based on concrete stress field was proposed to predict end concrete cover separation in RC beams strengthened with external FRP reinforcement. First, with the introduction of dowel action of steel and FRP reinforcement and the induced concrete splitting, which are the critical factors to reflect the interaction between concrete, steel and FRP reinforcement, the geometrical relationships of stress field for concrete were established through proper simplifications to configuration and mechanical state of cracked concrete block. Then, to assess the cracking angle and the correlated failure strength of concrete cover separation, the effective compressive strength of concrete in stress field was finely identified by incorporating the influential but prone to be neglected factors such as transverse strain, cracking status of surrounding concrete, and arrangement of internal steel reinforcement and external FRP reinforcement. Subsequently, an extended plane-section analytical approach, in which the components of effective tensile strain of FRP induced due to flexural and shear actions are both comprehensively considered according to the detailed location of concrete cover separation, is proposed to evaluate the carrying capacity of strengthened RC beams in a simple process. Finally, an excellent agreement between the predictions and experimental results was obtained to validate the proposed analytical approach; furthermore, the discussions and suggestions about the parameters concerned in the approach were proposed.

The detailed conclusions of this study can be drawn as follows:

By contrast with the conventional analytical models of concrete cover separation, merely focusing on the local response of concrete around FRP reinforcement, the proposed analytical approach based on stress field could comprehensively and properly consider the influence of the interaction between concrete, steel and FRP reinforcement, cracking status of surrounding concrete and its softening effect, and arrangement details of internal steel reinforcement and external FRP reinforcement on concrete cover separation; it is suited for practical design and analysis, and can be extended for external reinforcement with other composite materials.

Dowel action of steel and FRP reinforcement, and the induced concrete splitting are the critical factors to establish the geometrical relationships of stress field for cracked concrete block and to derive the failure strength of concrete cover separation, and cannot be neglected.

The assumption of the lower bound of splitting crack width of 0.015 $\emptyset_{s}$ could lead to the accurate prediction of the lower bound of cracking angle of stress field for concrete block.

The shear component in the effective tensile strain of FRP reinforcement cannot be ignored and was efficiently considered in the analysis of carrying capacity of strengthened RC beams that failed in end concrete cover separation.

Crack spacing has a great effect on the assessment of carrying capacity of strengthened RC beams and, consequently, the coefficient incorporating the amplification factor of average crack spacing ζ of 1.20 has been suggested to modify the effective strain of FRP. 

The proposed analytical approach obtained the satisfactory and conservatively safe prediction of the experimental results; by contrast, the commonly used concrete tooth model overestimated the failure strength of concrete cover separation, and generated unsafe prediction of carrying capacity of strengthened RC beams.

## Figures and Tables

**Figure 1 polymers-14-00988-f001:**
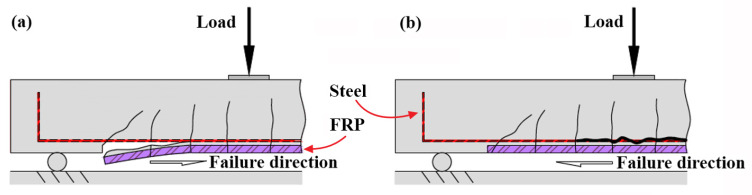
Location of concrete cover separation occurring at FRP-strengthened RC beams: (**a**) FRP reinforcement end; and (**b**) the maximum moment region.

**Figure 2 polymers-14-00988-f002:**
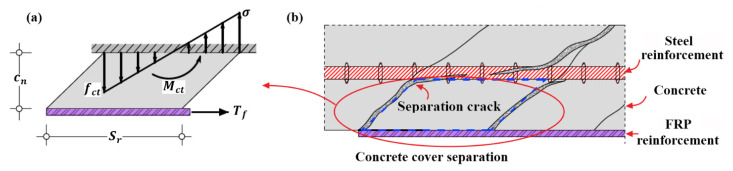
Concrete tooth model for end concrete cover separation: (**a**) mechanical model; and (**b**) concrete cover separation.

**Figure 3 polymers-14-00988-f003:**
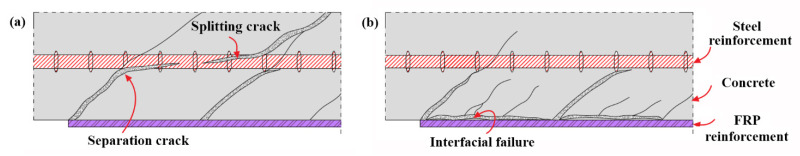
Schematical crack configurations: (**a**) cover separation; and (**b**) interfacial debonding.

**Figure 4 polymers-14-00988-f004:**
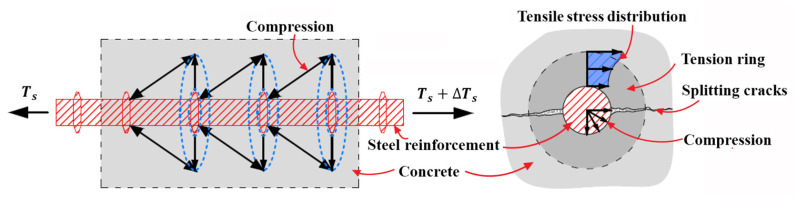
Tensile stress distributed in concrete and splitting cracks induced by bond action of steel reinforcement.

**Figure 5 polymers-14-00988-f005:**
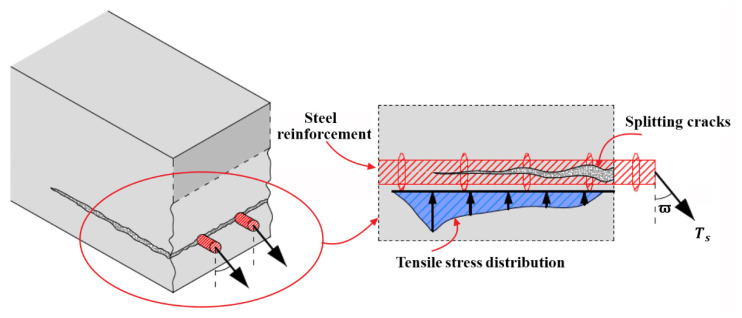
Tensile stress distributed in concrete and splitting cracks induced by dowel action of steel reinforcement.

**Figure 6 polymers-14-00988-f006:**
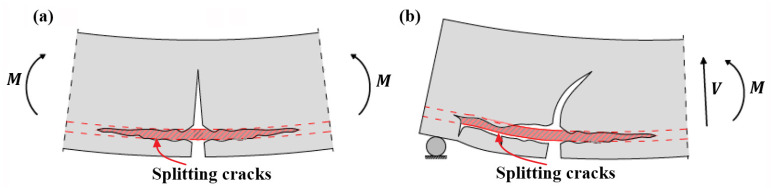
Typical formation locations of splitting cracks: (**a**) the maximum moment region; and (**b**) the end of RC beams.

**Figure 7 polymers-14-00988-f007:**
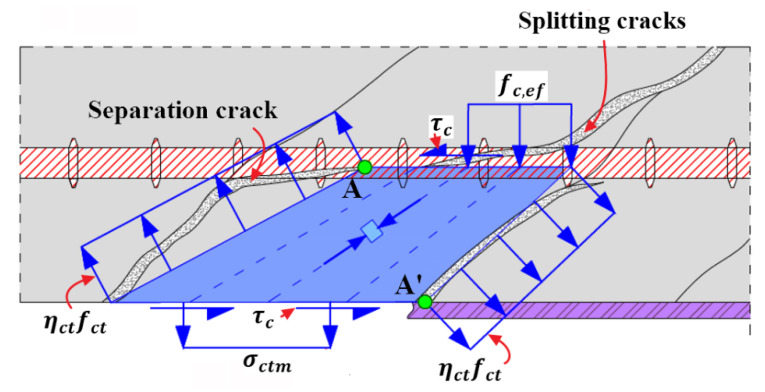
Schematical stress field for cracked concrete block at the end of FRP reinforcement.

**Figure 8 polymers-14-00988-f008:**
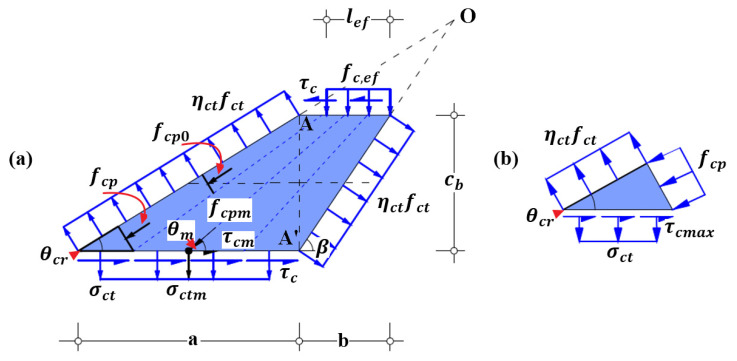
Stress field approach for predicting end concrete cover separation: (**a**) standardized stress field for cracked concrete block; and (**b**) the end of micro concrete strut.

**Figure 9 polymers-14-00988-f009:**
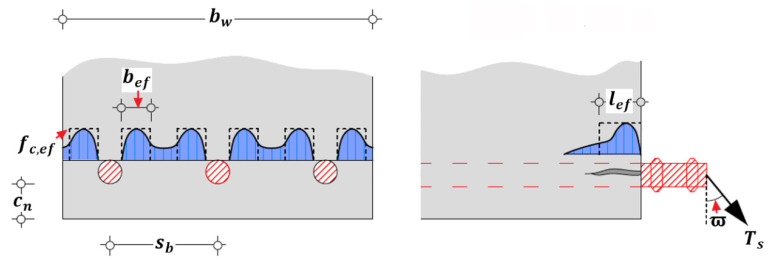
Calculation of dowel action of steel reinforcement [49,54,55,56].

**Figure 10 polymers-14-00988-f010:**
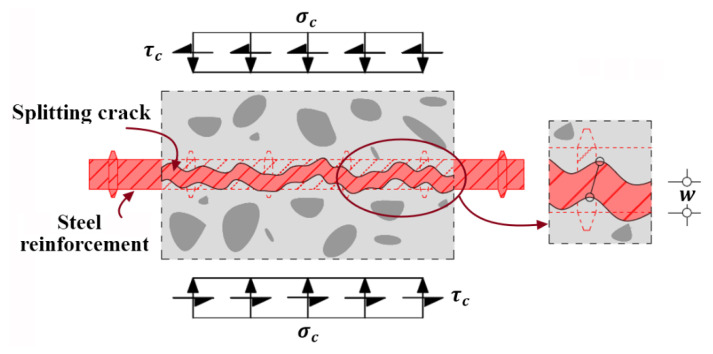
Details of cracked concrete with splitting crack.

**Figure 11 polymers-14-00988-f011:**
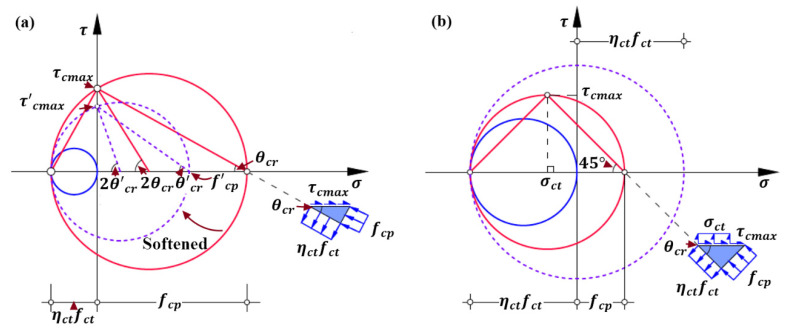
Mohr’s circle for the stress conditions of the micro concrete element at the end of FRP reinforcement: (**a**) Case 1; and (**b**) Case 2.

**Figure 12 polymers-14-00988-f012:**
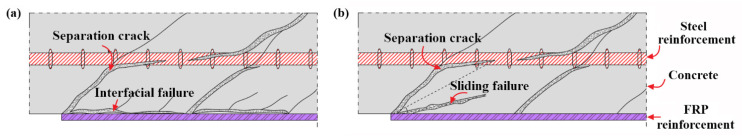
Cracking characteristics of mixed mode of end debonding failure: (**a**) mixed mode of interfacial debonding and cover separation; and (**b**) mixed mode of cover separation and sliding failure.

**Figure 13 polymers-14-00988-f013:**
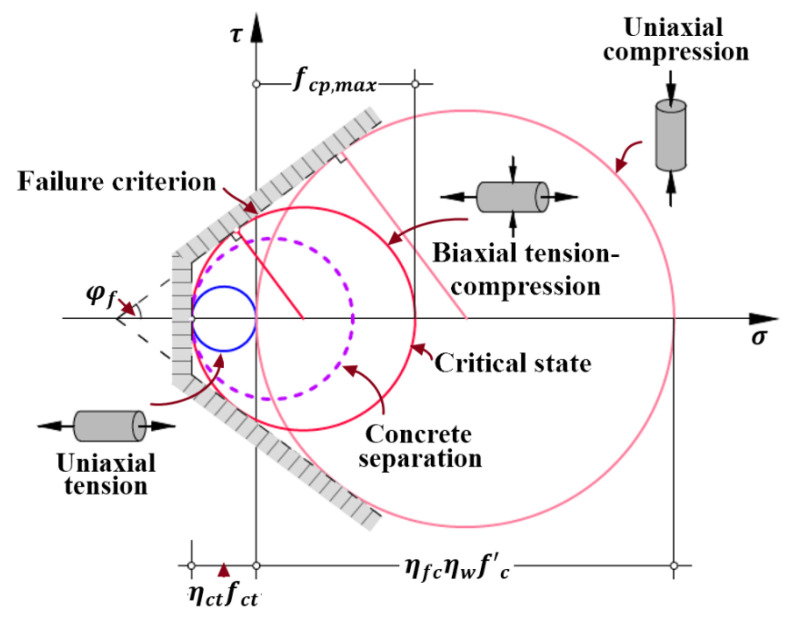
Mohr’s circle for the critical stress conditions of the micro concrete element at the end of FRP reinforcement with mixed mode of cover separation and sliding failure.

**Figure 14 polymers-14-00988-f014:**
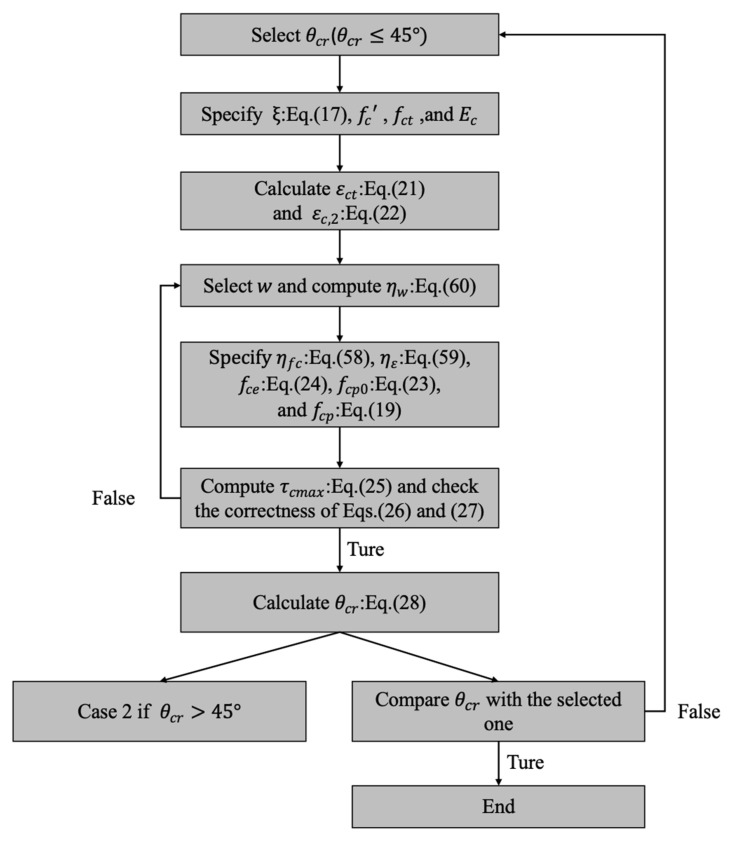
Analysis flowchart to estimate cracking angle $\theta_{cr}$.

**Figure 15 polymers-14-00988-f015:**
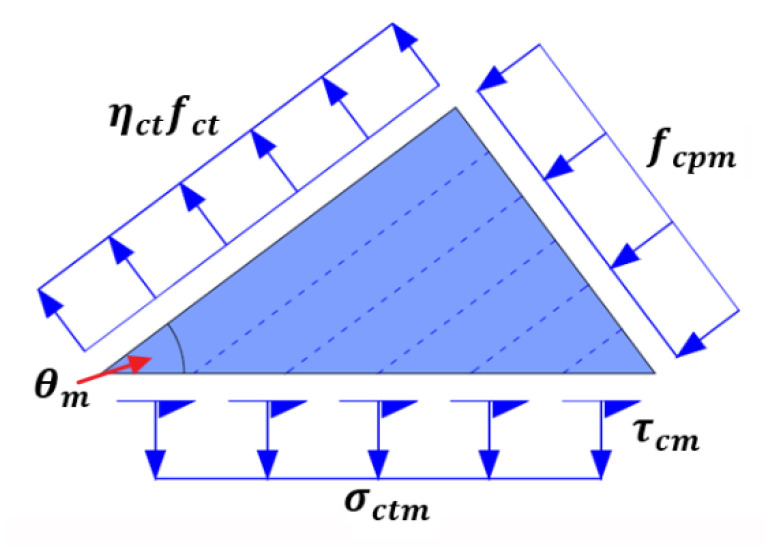
The average stress field for cracked concrete block.

**Figure 16 polymers-14-00988-f016:**
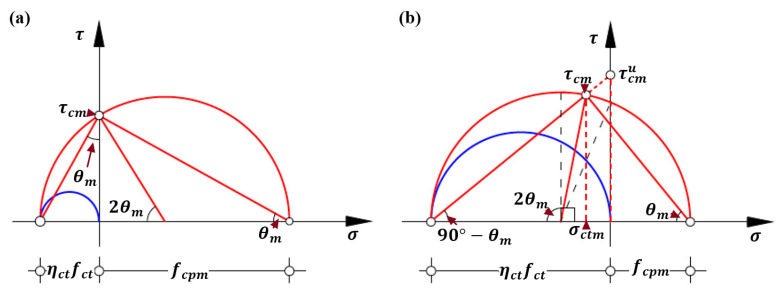
Mohr’s circle for the average stress field of cracked concrete block: (**a**) Case 1; and (**b**) Case 2.

**Figure 17 polymers-14-00988-f017:**
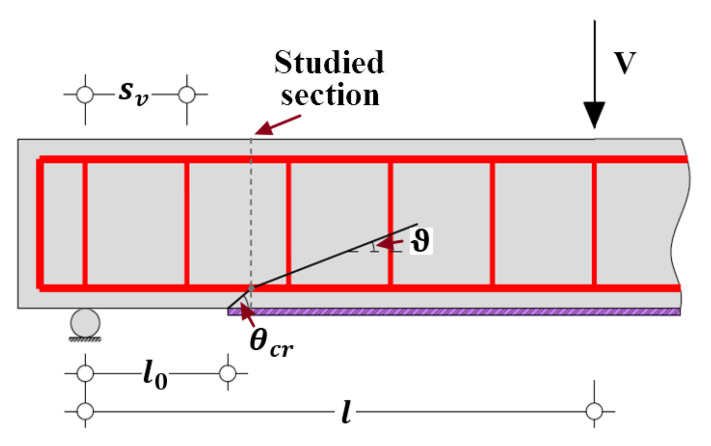
Geometries and configurations of FRP-strengthened RC beams.

**Figure 18 polymers-14-00988-f018:**
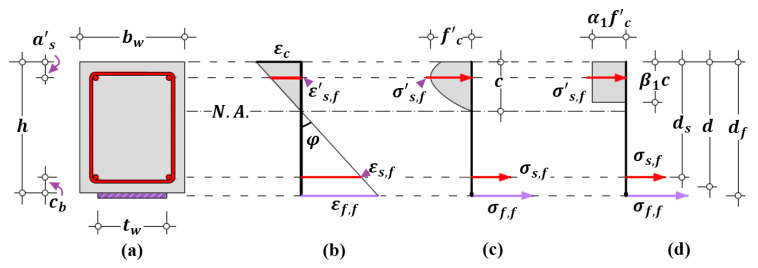
Geometries of beam section and profiles of strain and stress under flexural action: (**a**) beam cross-section; (**b**) strain distribution; (**c**) stress distribution; and (**d**) simplified stress distribution. (Note: N.A.= neutral axis).

**Figure 19 polymers-14-00988-f019:**
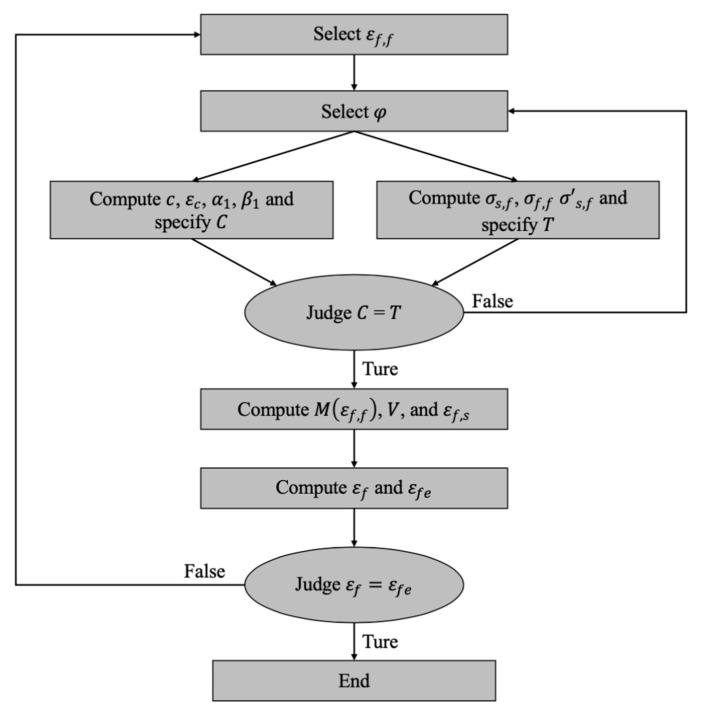
Analysis flowchart to assess the carrying capacity of strengthened RC beams.

**Figure 20 polymers-14-00988-f020:**
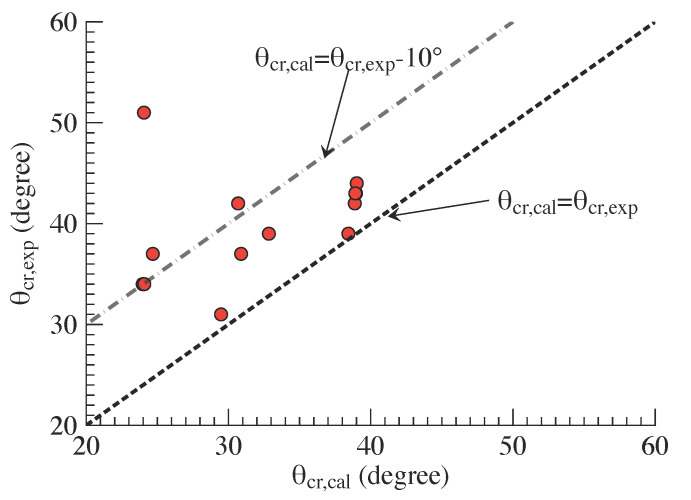
Comparisons between the predictions by the presented model and experimental results of cracking angle.

**Figure 21 polymers-14-00988-f021:**
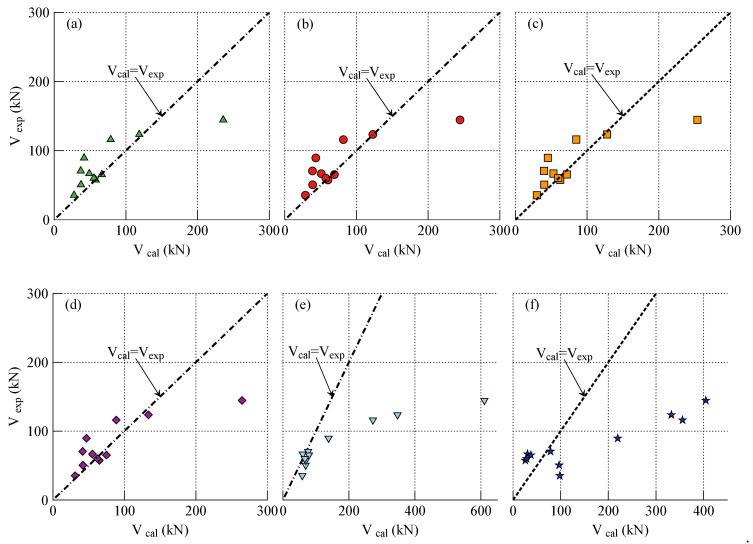
Comparisons between the predictions by the presented model, concrete tooth model, and experimental results of carrying capacity of FRP-strengthened beams, and the specification of amplification coefficient of average crack spacing ζ: (**a**) ζ=1.35; (**b**) ζ=1.30; (**c**); (**d**) ζ=1.20; (**e**) without modification; and (**f**) concrete tooth model.

**Table 1 polymers-14-00988-t001:** Geometries and material properties of beam specimens.

Reference	Specimen	Geometries						Mechanical Properties of Concrete		
		$l_{0}\;(\mathrm{mm})$	*l* (mm)	*h* (mm)	$b_{w}\;(\mathrm{mm})$	$c_{b}\;(\mathrm{mm})$	$a{’}_{s}\;(\mathrm{mm})$	$f{’}_{c}\;(\mathrm{MPa})$	$f_{ct}\;(\mathrm{MPa})$	$E_{c}\;(\mathrm{GPa})$
Smith et al. [9]	1B	25	500	250	205	45	45	31.5	2.4	23.3
	2B	125	500	250	205	45	45	48.6	3.6	28.8
	3B	50	500	250	205	45	45	45.3	3.2	29.0
	6B	75	500	250	205	45	45	41.0	2.9	29.4
Esfahani et al. [14]	B3	100	600	200	166	34	25	25.2	2.6	23.7
Yao et al. [16]	CS-L3-B	50	500	253	217	36	35	26.3	3.5	27.2
	CS-W100-B	50	500	254	214	41	35	30.2	3.3	24.3
	CP-B	50	500	253	218	35	35	26.2	3.8	27.4
Sabzi et al. [22]	5D18-F25-G	150	800	300	251	49	41	25.0	2.6	23.7
	5D10-F25-G	150	800	300	267	33	39	25.0	2.6	23.7
Sabzi et al. [23]	2D22-NSG-G	150	800	300	251	49	44	25.0	2.6	23.7
	5D14-NSC-G	150	800	300	255	45	44	25.0	2.6	23.7
Pham et al. [27]	E1a	150	700	260	220	40	52	53.7	4.3	34.7

**Table 2 polymers-14-00988-t002:** Mechanical and geometrical properties of steel reinforcement and FRP reinforcement.

Specimen	Tensile Steel Longitudinal Reinforcement				Compressive Steel Longitudinal Reinforcement			Steel Transverse Reinforcement				Geometries of FRP		Mechanical Properties of FRP	
	$n_{b}$	$\emptyset_{s}\;(\mathrm{mm})$	$E_{s}\;(\mathrm{GPa})$	$f_{sy}\;(\mathrm{MPa})$	$A{’}_{s}\;({\mathrm{mm}}^{2})$	$E{’}_{s}\;(\mathrm{GPa})$	$f{’}_{sy}\;(\mathrm{MPa})$	$A_{sv}\;({\mathrm{mm}}^{2})$	$s_{v}\;(\mathrm{mm})$	$E_{sv}\;(\mathrm{GPa})$	$f_{yv}\;(\mathrm{MPa})$	$b_{f}\;(\mathrm{mm})$	$t_{f}\;(\mathrm{mm})$	$E_{f}\;(\mathrm{GPa})$	$f_{fu}\;(\mathrm{MPa})$
1B	2	10	207	506	157.1	207	506	157.1	100	207	506	150	1.77	271	3720
2B	2	10	207	506	157.1	207	506	157.1	100	207	506	148	1.70	271	3720
3B	2	10	207	506	157.1	207	506	157.1	100	207	506	147	1.87	257	4591
6B	2	10	207	506	157.1	207	506	157.1	100	207	506	145	1.81	257	4591
B3	2	12	200	400	157.1	200	365	100.5	80	200	350	150	0.35	237	2845
CS-L3-B	2	10	199	536	157.1	199	536	157.1	100	199	536	148	2.63	256	4114
CS-W100-B	2	10	199	536	157.1	199	536	157.1	100	199	536	100	1.95	256	4114
CP	2	10	199	536	157.1	199	536	157.1	100	199	536	148	1.20	165	2800
5D18-F25-G	5	18	223	367	226.2	210	412	157.1	80	190	462	160	0.17	240	3600
5D10-F25-G	5	10	190	462	226.2	210	412	100.5	120	190	462	160	0.17	240	3600
2D22-NSG-G	2	22	204	376	226.2	210	412	100.5	100	190	462	160	0.17	240	4950
5D14-NSC-G	5	14	205	423	226.2	210	412	100.5	100	190	462	160	0.17	240	4950
E1a	3	12	205	551	226.2	205	551	157.1	100	204	334	100	1.06	209	3900

**Table 3 polymers-14-00988-t003:** Comparisons between the calculational results and experimental results about cracking angle and carrying capacity of the strengthened RC beams.

Specimen	$\theta_{cr,exp}(\mathrm{degree})$	$\theta_{cr,cal}/\theta_{cr,exp}$	$V_{exp}(\mathrm{kN})$	$V_{cal}^{1.35}/V_{exp}$	$V_{cal}^{1.3}/V_{exp}$	$V_{cal}^{1.25}/V_{exp}$	$V_{cal}^{1.2}/V_{exp}$	$V_{cal}^{0}/V_{exp}$	$V_{cal}^{ctm}/V_{exp}$
1B	37	0.67	66.80	0.73	0.76	0.79	0.83	0.90	0.46
2B	51	0.47	57.60	1.01	1.04	1.09	1.13	1.20	0.45
3B	34	0.71	65.40	1.02	1.06	1.10	1.14	1.21	0.57
6B	34	0.71	60.20	0.92	0.95	0.99	1.03	1.10	0.47
B3	31	0.95	35.47	0.78	0.81	0.84	0.88	1.65	2.77
CS-L3-B	37	0.83	-	-	-	-	-	-	-
CS-W100-B	39	0.99	-	-	-	-	-	-	-
CP	42	0.73	50.70	0.74	0.77	0.80	0.83	1.36	1.91
5D18-F25-G	42	0.93	144.50	1.63	1.69	1.76	1.83	4.22	2.80
5D10-F25-G	43	0.91	89.50	0.47	0.48	0.51	0.53	1.54	2.45
2D22-NSG-G	44	0.89	116.00	0.68	0.70	0.73	0.76	2.35	3.07
5D14-NSC-G	43	0.91	123.50	0.96	0.99	1.03	1.08	2.81	2.69
E1a	39	0.84	70.70	0.52	0.54	0.57	0.59	1.08	1.11
Average		0.81		0.86	0.89	0.93	0.97	1.77	1.71
Standard deviation		0.15		0.32	0.33	0.34	0.35	1.00	1.10

## Data Availability

No new data were created or analyzed in this study. Data sharing is not applicable to this article.

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
