# Peer review of "Stress Field Approach for Prediction of End Concrete Cover Separation in RC Beams Strengthened with FRP Reinforcement"

_polymers, 2022, doi:10.3390/polym14050988_

Round 1

Reviewer 1 Report

  1. The authors must include the motivation behind the work.
  2.  The novelty of the work must be incorporated at the end of the Introduction.
  3. The Introduction section does not contains figures.
  4. The theory behind the work must be included separately in materials and methods.
  5. The paper is written like a book, and should be modified as a scientific paper.
  6. The length of the paper must be reduced.
  7. Many of the equations are used without quoting the source.
  8. The FRP reinforcement must be discussed properly.
  9. The discussion must concentrate more on stress field approach.
  10. . Improve the conclusions.
  11. Polymer contents and its details must be explained properly.
  12. How this manuscript is going to be considered for Polymers.
  13. Improve the conclusions.
  14. Provide more recent references.

Reviewer 2 Report

Although the article proposed an analytical model to predict the failure strength of concrete cover separation, the well-known models did not compare nor mention. Furthermore, the following article, https://doi.org/10.3151/jact.2.419, compared between the experimental results and three of these well-known models. This comparison should be extended in the present manuscript to be as a verification of the proposed model. Therefore, this manuscript needs essential modifications as follows:

  • The introduction must include the well-known related models with an explanation of each model’s hypotheses, advantages, and disadvantages.
  • A comparison with these models should be added.

Reviewer 3 Report

Submitted manuscript deals with an analytical approach based on concrete stress field to predict end concrete cover separation in RC beams strengthened with external FRP reinforcement. Dowel action of steel and FRP reinforcement, concrete splitting, mechanical state of cracked concrete block are merged in a comprehensive model. Text is extremely long and difficult to follow (please consider to put some explanations in separate appendices, to make the main text more readable). Howsoever the manuscript provides a proposal that is worth of presentation. English is in general readable, however there are many typos to be corrected.

I suggest to stress since the beginning that the model is based on point stress evaluations (local stress) to derive a global failure for the cover region, hence it should be clarified that redistribution and plasticity could improve this behaviour, hence this is a safe side evaluation (potentially an experimental validation could provide experimental calibration of the final equations to overcome this potential overconservativeness).

Furthermore, the end debonding failure is a failure mode not strictly related to flexural failure if the FRP has enough bond length to provide the effective stress in FRP at maximum bending moment section. Howsoever the cover separation related to end debonding is a failure mode that need to be prevented. This clarification is required since the beginning, otherwise, it seems that effective strain at end debonding is the strain of FRP at the maximum bending moment section (two sections are potentially far away). This is particularly required with reference to section 4.

I suggest to improve the introduction accounting also for different FRP flexural strengthening systems, not depending on bond; e.g. with mechanical fasteners, where the benefit is the prevention of debonding, but the drawback is the need to drill and insert fasteners into the concrete cover and soffit of beams (for instance, among others, authors could consider https://doi.org/10.1016/j.compstruct.2011.03.003 ). As further examples, end anchorage systems should be cited, too (for instance, among others, authors could consider https://doi.org/10.1007/s40069-013-0029-0 ).

Looking at figure 2 authors are encouraged to consider that the concrete cover is a tooth/cantilever that is not only loaded in bending, but loaded by combined bending and axial compression, where the axial compression is the component of the traction T in the diagonal direction.

Steps described at lines 502, and following lines, are not clear (numbering is missing) and I suggest to provide a flowchart to clearly identify each step of the described procedure. Cited steps 6 and 9 are not defined in the procedure in its present form.

Similarly, steps described at lines 697, and following lines, are not clear (numbering is missing) and I suggest to provide a flowchart to clearly identify each step of the described procedure. Cited steps 2-5 and 1-8 are not defined in the procedure in its present form.

Conclusions are long, however the comparisons with commonly used concrete tooth model cited at the end of conclusions are missing, while it would be interesting to compare the proposed model with previous, available models.

Round 2

Reviewer 2 Report

The authors have successfully addressed all my comments.  Therefore, I recommend the publication of this manuscript.